# Tissue-resident macrophages are major tumor-associated macrophage resources, contributing to early TNBC development, recurrence, and metastases

Ryuichiro Hirano[1], Koki Okamoto[1], Miyu Shinke[1], Marika Sato[1], Shigeaki Watanabe[1], Hitomi Watanabe[2], Gen Kondoh[2], Tetsuya Kadonosono[1] & Shinae Kizaka-Kondoh [1✉]

Triple-negative breast cancer (TNBC) is an aggressive and highly heterogenous disease with no well-defined therapeutic targets. Treatment options are thus limited and mortality is significantly higher compared with other breast cancer subtypes. Mammary gland tissue-resident macrophages (MGTRMs) are found to be the most abundant stromal cells in early TNBC before angiogenesis. We therefore aimed to explore novel therapeutic approaches for TNBC by focusing on MGTRMs. Local depletion of MGTRMs in mammary gland fat pads the day before TNBC cell transplantation significantly reduced tumor growth and tumor-associated macrophage (TAM) infiltration in mice. Furthermore, local depletion of MGTRMs at the site of TNBC resection markedly reduced recurrence and distant metastases, and improved chemotherapy outcomes. This study demonstrates that MGTRMs are a major TAM resource and play pivotal roles in the growth and malignant progression of TNBC. The results highlight a possible novel anti-cancer approach targeting tissue-resident macrophages.

[1] School of Life Science and Technology, Tokyo Institute of Technology, Yokohama 226-8501, Japan. [2] Institute for Life and Medical Sciences, Kyoto University, Sakyo, Kyoto 606-8507, Japan. ✉email: skondoh@bio.titech.ac.jp

Triple-negative breast cancer (TNBC) is an aggressive and highly heterogenous breast cancer subtype that lacks diagnostic and therapeutic markers, such as estrogen receptor, progesterone receptor, and human epidermal growth factor receptor 2[1,2]. There is currently no specific treatment for TNBC, which has a higher risk of early local recurrence after surgery compared with other breast cancer subtypes, and is often associated with distant metastasis[3–5]. TNBC thus has the poorest prognosis of all breast cancers[6], and novel treatment strategies are urgently required to improve the outcomes of patients with TNBC.

Current trends and challenges in cancer treatment have focus on tumor-infiltrating stromal cells, especially immune cells associated with the malignant progression of cancer[7]. Among them, tumor-associated macrophages (TAMs) are the most abundant immune cells in many tumors, and contribute to creating a microenvironment that promotes immunosuppression, metastasis/invasion, angiogenesis, and drug resistance[8–11]. Tumor infiltration by TAM has been correlated with poor patient outcomes[12]. Although various studies have investigated TAM-specific targeting, a successful approach has not been found, as all the approaches used to date have demonstrated limitations[8,10,11]. The major limitations affecting TAM-specific targeting include diversity among macrophage populations[13], which makes it difficult to target all pro-tumorigenic TAMs without affecting macrophages that have anti-tumorigenic functions and play a role in immune surveillance and tissue homeostasis[14].

TAMs can differentiate from circulating monocytes that infiltrate tumors via the blood vessels, but may also derived from tissue-resident macrophages (TRMs) that infiltrate tumors directly from the surrounding tissues. TRMs are known to contribute to tumor progression in some tumor types, such as brain tumors, pancreatic cancer, and lung cancer, through functions different from those of TAMs derived from bone marrow-derived monocytes[15–20]. TRMs have several origins: yolk sac macrophages, fetal liver monocytes, or adult bone-marrow monocytes and locally self-replicate independently of adult hematopoiesis[21]. TRMs have tissue-specific characteristics[22] and contribute to biological homeostasis, such as tissue development and repair, response to infection, and inflammation resolution[14,23,24]. Most mammary gland tissue-specific TRMs (MGTRMs) are derived from the yolk sac and fetal liver[25] and play a role in mammary tissue development and homeostasis by interacting with mammary epithelial cells[26–28]. MGTRMs, with distinct functions from myeloid-derived TAMs, may infiltrate into breast cancers and contribute to their growth and malignant progression. Although some studies have investigated the involvement of MGTRMs in breast cancers[29–31], the significance of MGTRMs as a therapeutic target in breast cancer has remained unexplored.

Here, we investigate the function of MGTRMs during early TNBC development and reveals that MGTRMs are a major TAM resource and play pivotal roles in TNBC growth and malignant progression. Furthermore, we investigate the effect of targeting MGTRMs on early local recurrence after surgery and subsequent distant metastasis. The results suggest that targeting MGTRMs may be a promising therapeutic strategy for improving the prognosis of patients with TNBC. Our study thus provides significant insight into the potential of TRMs as promising therapeutic targets for many cancers.

## Results

**MGTRMs are the major stromal cell population in TNBC during early tumor development.** To investigate the stromal cells infiltrating into TNBC from the surrounding tissues, the C57BL/6 J (B6) mouse-derived TNBC cell line E0771[32] was injected into the fourth mammary gland fat pad (MGFP) of B6 mice, and the timing of tumor angiogenesis was determined. Immunohistochemical analysis showed the presence of CD31[+] endothelial cells in E0771 tumors at 4 days after transplantation, but not 3 days after transplantation (Fig. 1a left). The similar results were obtained using the BALB/c mouse-derived TNBC cell line 4T1[33] (Fig. 1a right). These findings indicate that tumor blood vessels began to form 4 days after transplantation in this model, and tumor stromal cells observed before this had therefore infiltrated directly from the surrounding tissue, not through blood vessels.

We quantitatively assessed tissue-resident cell infiltration in B6 green fluorescent protein (GFP)-transgenic (Tg) mice that constitutively express GFP in cells throughout the body. Immunohistochemical analysis of E0771 tumors 3 days after transplantation showed that 56% of GFP[+] stromal cells expressed the macrophage marker F4/80, and 10% expressed the fibroblast marker α-smooth muscle actin (α-SMA) (Fig. 1b). T cells and dendritic cells known to be present in MGFP[34,35] were also investigated and showed very limited infiltration into TNBC 3 days after transplantation (Supplementary Fig. 1). F4/80[+] macrophages were also detected in 4T1 tumors 3 days after transplantation (Fig. 1c).

About a half (42–49%) of the F4/80[+] macrophages were positive for folate receptor beta (FOLR2) and mannose receptor C-Type 1 (MRC1/CD206) (Fig. 1d, e), which were co-expressed on a MGTRM subset[31]. As we expected, the F4/80[+] macrophages did not express CADM1, a monocyte-derived macrophage marker[31] (Fig. 1f). These results indicate that the F4/80-positive macrophages infiltrating early TNBC are MGTRMs, with the FOLR2[+] subset accounting for half. Since the other half of MGTRMs has not been identified, we decided to investigate the entire MGTRMs for their contribution to promoting TNBC growth.

**MGTRMs promote TNBC cell proliferation in vitro.** We investigated the function of MGTRMs in the growth of TNBC by examining the effect of MGFP on TNBC cell proliferation in vitro. TNBC cell proliferation was significantly increased by co-culture with MGFP, with a filter in between to prevent direct contact (Fig. 2a), revealing that secreted factors in the co-culture medium promoted the proliferation of TNBC cells. The treatment of MGFP with clodronate liposomes (CL), which induces apoptosis in macrophages[36], significantly reduced the proliferation of TNBC cells, whereas treatment of MGFP with control phosphate-buffered saline (PBS) liposomes (PL) did not affect TNBC cell proliferation (Fig. 2a). These results suggested that MGTRMs were critically involved in the growth of TNBC. Stimulation of TNBC cell proliferation by MGTRMs was confirmed using a co-culture assay with MGTRMs isolated form MGFP in direct contact conditions (Fig. 2b and Supplementary Fig. 2). Treatment of TNBC cells with CL in vitro had no effect on TNBC cell proliferation (Fig. 2c). These results suggest that MGTRMs may promote TNBC growth in vivo.

**Depletion of MGTRMs suppresses early TNBC development in vivo.** To confirm the effect of MGTRMs on TNBC growth in vivo, we depleted MGTRMs in MGFPs by topical administration of CL to the fourth MGFP, and then investigated the growth of transplanted TNBC cells. As CL has been reported to efficiently deplete macrophages[26,37], a single topical administration of CL was shown to efficiently reduced (~86%) the number of MGTRMs from the MGFP of tumor-free mice 1 day after administration by flow cytometry (Fig. 3a and Supplementary Fig. 3a) and immunohistochemical analysis (Supplementary Fig. 3b). CL has been shown to target phagocytic cells[36,38], and

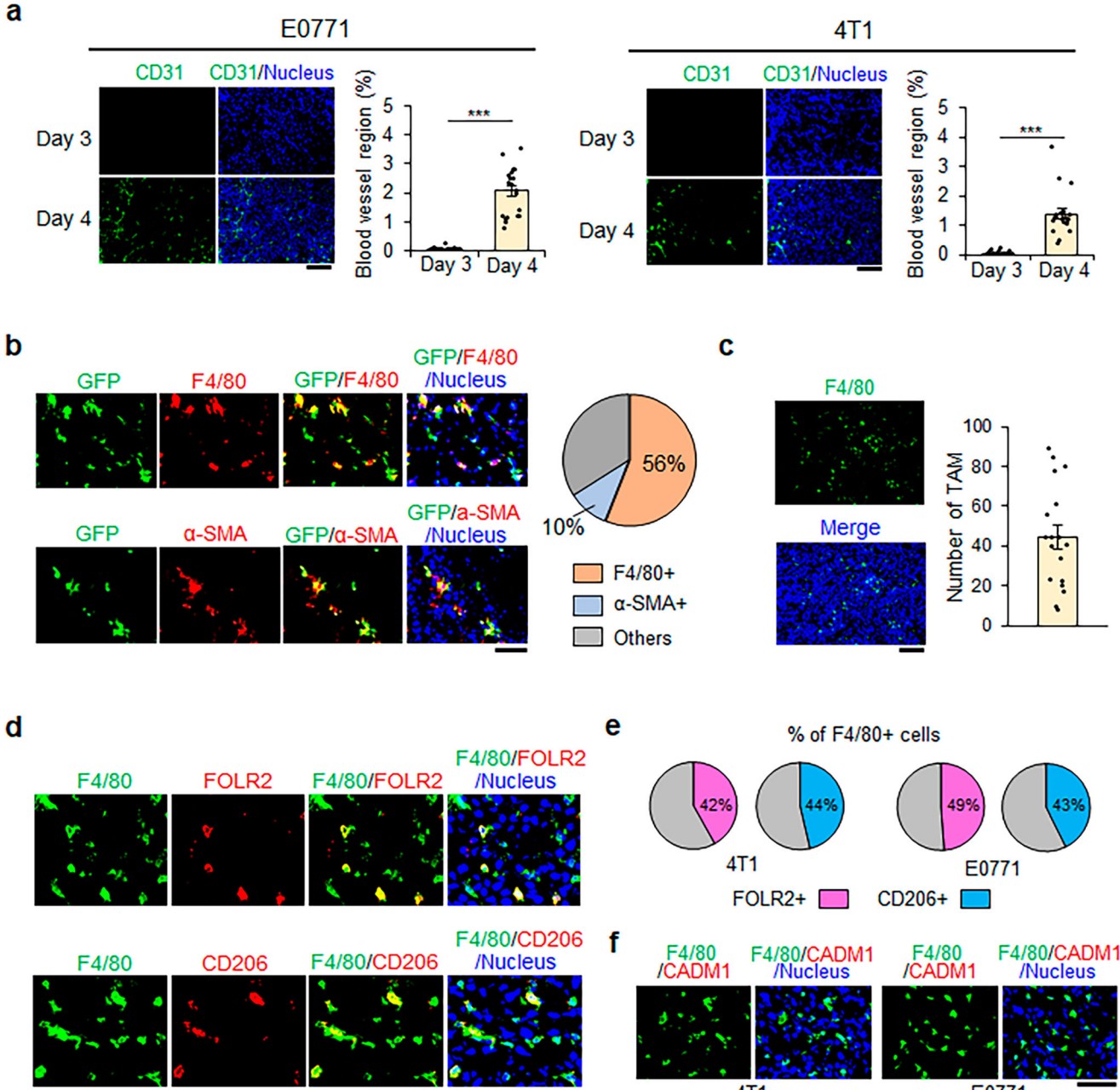

**Fig. 1 Infiltration of mammary gland tissue-resident macrophages (MGTRMs) in pre-angiogenic triple-negative breast cancer (TNBC).**
**a** Representative fluorescence images (left-hand panel) and average % of the CD31$^+$ area (right-hand panel) in tumors (1–2 mm in diameter) 3 and 4 days after the transplantation of indicated TNBC cells. $n = 20$. ***$p < 0.001$ ($t$-test). Scale bar = 100 μm. **b** Sections of E0771 tumors (~1 mm in diameter) in green fluorescent protein (GFP)-transgenic mice 3 days after transplantation. Representative fluorescence images of GFP$^+$, F4/80$^+$, and α-smooth muscle actin (SMA)$^+$ cells (left) and average % of F4/80$^+$ and α-SMA$^+$ cells among GFP$^+$ cells (right). $n = 20$. Scale bar = 50 μm. **c** Representative image (left) and average number of F4/80$^+$ cells in a field (right) of 4T1 tumors (~1 mm in diameter) 3 days after transplantation. $n = 18$. Scale bar = 100 μm.
**d** Sections of 4T1 tumors (~1 mm in diameter) in BALB/c mice 3 days after transplantation. Representative fluorescence images of F4/80$^+$, FOLR2$^+$, and CD206$^+$ cells. Scale bar = 50 μm. **e** Average % of FOLR2$^+$ and CD206$^+$ cells among F4/80$^+$ MGTRMs. $n = 15$. **f** Sections of TNBC (~1 mm in diameter) 3 days after transplantation. Representative fluorescence images of F4/80$^+$ and CADM1$^+$ cells. Scale bar = 50 μm. **a**, **c** Error bars indicate standard error of the mean (s.e.m.).

neutrophils have also been identified as professional phagocytic cells in MGFPs[39]. We therefore examined the effect of CL on neutrophils. The number of neutrophils in untreated MGFP was low, with no change 1 day after CL administration (Supplementary Fig. 3c), indicating that CL had little or no effect on neutrophils. Furthermore, since dendritic cells are known to present in MGFPs[35] and also killed by CL[40], we examined

abundance of dendritic cells in MGFP before CL treatment and found that they were very few (0.59%) in MGFP (Supplementary Fig. 3d), consistent with very little infiltration of CD11c$^+$ cells into early TNBC (Supplementary Fig. 1). Therefore, mammary gland tissue-resident dendritic cells appear to infiltrate TNBC in fairly low numbers and have a very limited effect on TNBC growth.

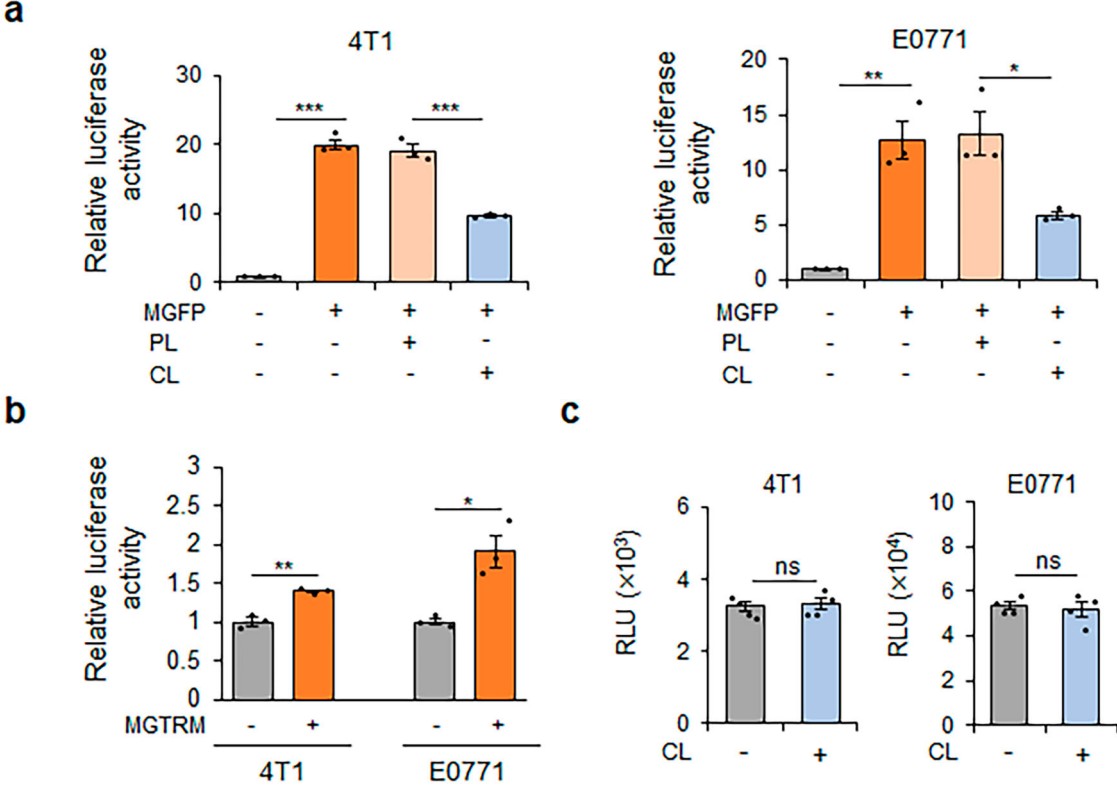

**Fig. 2 TNBC proliferation is promoted by mammary gland fat pad (MGFP) and MGTRMs in vitro. a** 4T1 and E0771 cells carrying luciferase reporter genes were co-cultured with MGFP treated with PBS liposomes (PL) or clodronate liposomes (CL) for 3 days. Luciferase activity of TNBC cells was normalized by the luciferase activity of corresponding monocultured TNBC cells. $n = 3$. *$p < 0.05$, **$p < 0.01$, ***$p < 0.001$ ($t$-test). **b** 4T1 and E0771 cells carrying luciferase reporter genes were co-cultured with MGTRMs for 3 days. Luciferase activity of TNBC cells was normalized by the luciferase activity of corresponding monocultured TNBC cells. $n = 3$. *$p < 0.05$, **$p < 0.01$ ($t$-test). **c** Luciferase activity of 4T1/Fluc and E0771/mKO2-luc2 cells cultured with ($+$) or without ($-$) CL for 3 days was measured and their relative light units (RLU) are shown. $n = 4$. ns not significant ($t$-test). **a–c** Error bars indicate s.e.m.

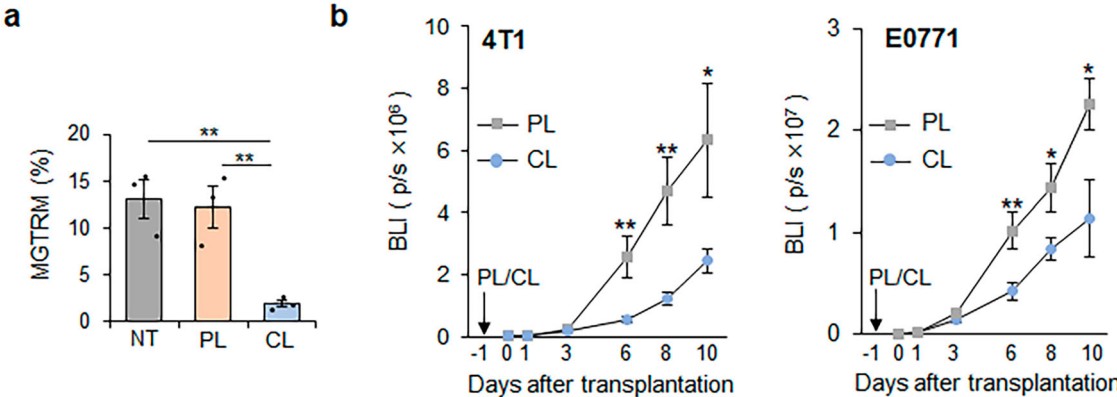

**Fig. 3 Effect of MGFP treatment with CL on TNBC growth. a** Flow cytometry analysis of CD11b$^+$F4/80$^+$ MGTRMs in BALB/c MGFP 1 day after PL or CL administration (Left). $n = 3$. NT: untreated. **$p < 0.001$ ($t$-test). **b** Bioluminescence intensity (BLI) of indicated TNBC transplanted 1 day after PL or CL administration at indicated days after transplantation. $n = 12$ for PL, $n = 14$ for CL. *$p < 0.05$, **$p < 0.01$ ($t$-test). **a, b** Error bars indicate s.e.m.

TNBC cells expressing a firefly luciferase reporter gene were transplanted into the MGFP 1 day after CL administration, and bioluminescence signals from the resulting TNBC were monitored over time. The early growth of TNBC in CL-treated MGFPs was significantly reduced compared with that in PL-treated MGFPs (Fig. 3b, Supplementary Fig. 3e). Since CL did not affect TNBC cell proliferation (Fig. 2c), we can conclude that the reduction in TNBC growth was mainly due to CL-mediated depletion of MGTRMs. Taken together, these results indicated that MGTRMs are primary stromal cells infiltrating early TNBC and significantly contributed to TNBC growth.

**MGTRMs are a major resource of TAMs and promote angiogenesis in TNBC.** Correlating well with the CL treatment effect on TNBC growth rate, the number of TAMs infiltrating TNBC was significantly lower in tumors in CL-treated MGFPs than those in PL-treated MGFPs by day 10 post-transplantation (Fig. 4a), supporting the hypothesis that MGTRMs function as TAMs and contributes to promoting TNBC growth. Given the multifunctionality of macrophages in general, in addition to the direct stimulation of TNBC cell proliferation (Fig. 2b), the indirect TNBC growth-promoting functions of MGTRMs were investigated. The vascular density of 4T1 tumors in CL-treated MGFPs

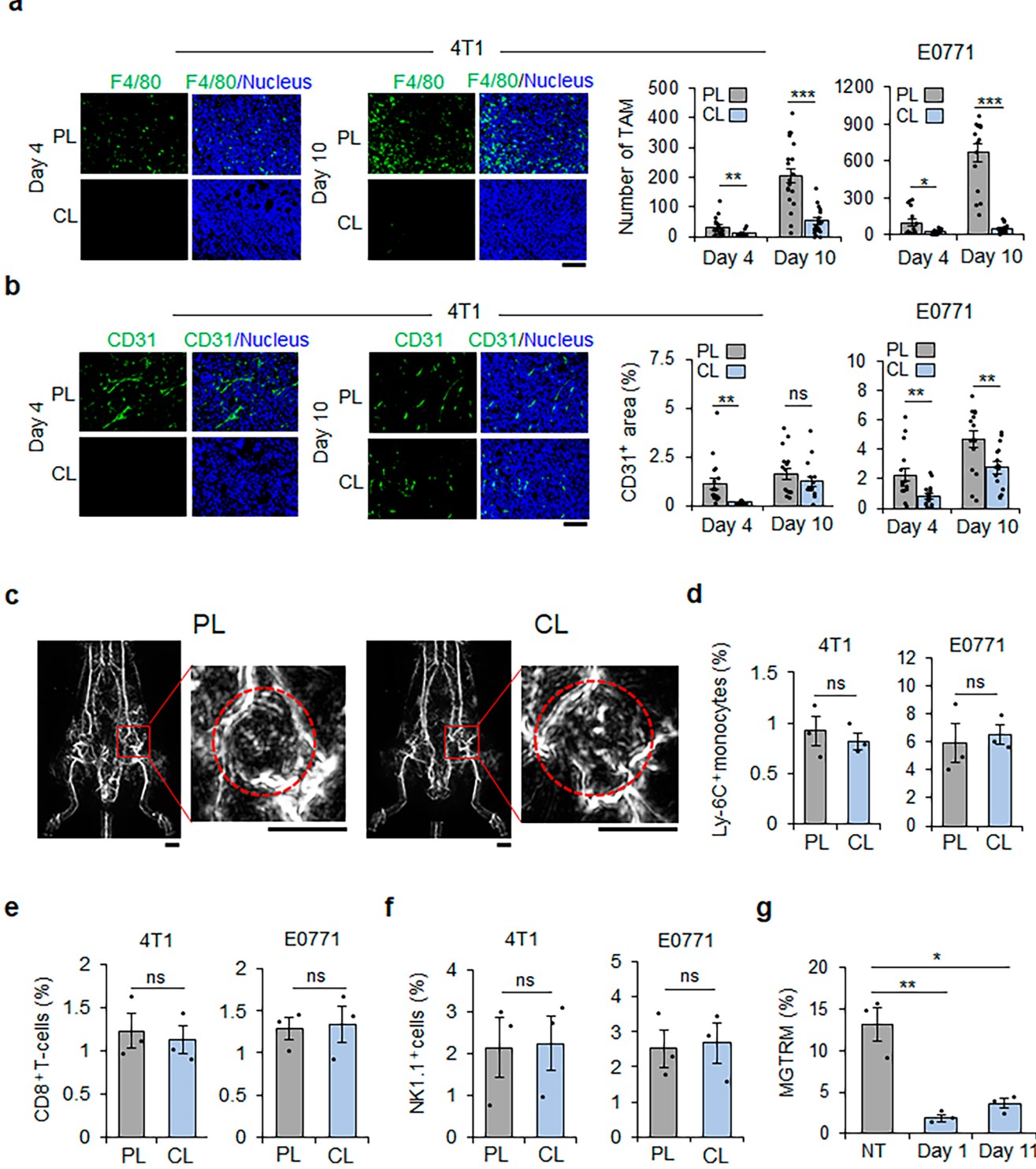

**Fig. 4 MGTRMs are a major resource of tumor-associated macrophages (TAMs) and contribute to TNBC angiogenesis. a** Representative fluorescence images of 4T1 tumor (left) and average number of F4/80+ cells in a field (right) in TNBC 4 and 10 days after transplantation. 4T1: $n = 20$, E0771: $n = 15$, *$p < 0.05$, **$p < 0.01$, ***$p < 0.001$ (t-test). Scale bar = 100 μm. **b** Representative fluorescence images of 4T1 tumor (left) and average % of the CD31+ area (right) in TNBC 4 and 10 days after transplantation. $n = 15$. ns, not significant, **$p < 0.01$ (t-test). Scale bar = 100 μm. **c** Photoacoustic images of blood vessels in mice 10 days after 4T1 cell transplantation in PL- and CL-treated MGFPs. The tumor areas indicated by red squares are enlarged in the right-hand panels. The red dotted circle indicates the tumor area. Scale bar = 5 mm. **d–f** Percentage of CD11b+Ly-6C+ monocytes, CD8+ T cells, and NK1.1+ cells among all tissue cells of TNBC 10 days after transplantation in PL- and CL-treated MGFP. $n = 3$. ns, not significant (t-test). **g** Flow cytometry analysis of CD11b+F4/80+ MGTRMs in tumor-free BALB/c MGFP 1 and 11 days after CL administration. $n = 3$. *$p < 0.05$, **$p < 0.01$ (t-test). **a**, **b**, **d–g** Error bars indicate s.e.m.

was significantly lower than that in PL-treated MGFPs on day 4, although there was no significant difference on day 10 (Fig. 4b). E0771 tumors showed significantly lower vascular density in CL-treated MGFPs compared with PL-treated MGFPs on day 10 as well as day 4 (Fig. 4b), suggesting that MGTRMs infiltrating TNBC also promote angiogenesis.

Photoacoustic imaging, which mainly uses the light absorption of hemoglobin present in the vasculature[41], confirmed the presence of erythrocytes in the blood vessels of TNBCs in both PL- and CL-treated MGFPs 10 days after 4T1 transplantation, revealing that the blood vessels were functional (Fig. 4c, Supplementary Movies 1, 2). This indicates that 4T1 tumors were ready for bone marrow-derived cell infiltration on day 10 with or without CL treatment. The numbers of bone marrow-derived Ly-6C$^+$ monocytes, CD8$^+$ T cells, and NK1.1$^+$ cells that infiltrated tumors on day 10 (corresponding to day 11 after CL/PL administration) were comparable between TNBC in CL- and PL-treated MGFPs, thus confirming that bone marrow-derived cells were able to infiltrate into tumors regardless of CL treatment (Fig. 4d–f). These findings indicate that CL treatment did not affect the infiltration of monocytes capable of differentiating into TAMs in early TNBC.

The low number of TAMs in TNBC in CL-treated MGFPs despite the formation of functional tumor vasculature on day 10 (Fig. 4a–c) suggested that MGTRMs were major TAM resources during the early development of TNBC. Accordingly, the duration of TAM reduction in TNBC in CL-treated MGFPs would be expected to correspond to the recovery period of MGTRMs in MGFPs after CL treatment. To verify this, the number of F4/80$^+$ cells in MGFPs treated with CL was examined by flow cytometry 1 day and 11 days after the topical administration of CL to tumor-free mice. The number of MGTRMs was significantly lower on both days compared with untreated controls (Fig. 4g and Supplementary Fig. 4), revealing slow recovery of MGTRMs after the single CL treatment of MGFPs. These results support the hypothesis that MGTRMs rather than cells of a bone marrow origin are the major resource for TAMs during early TNBC development.

**MGTRMs depletion significantly suppresses local recurrence and distant metastases of TNBC.** Given that MGTRM-derived TAMs contribute significantly to the early development and malignant progression of TNBC, local depletion of MGTRMs would be expected to suppress early postoperative recurrence and distance metastases, which are leading causes of the poor prognosis in TNBC patients[6]. To investigate this, we evaluated the effect of MGTRMs depletion by CL on TNBC recurrence using a local recurrence model, in which 95% of the tumor was removed when reached 1 cm in diameter[42]. To optimize the frequency of CL administration, PL and CL were administered topically to the tumor resection sites on the day of surgery (single), on the day of surgery and 2 days after surgery (twice), and for 4 consecutive days from the day of surgery (4 daily) (Fig. 5a), and the growth of recurrent tumors was then monitored by bioluminescence imaging. "Twice" and "4 daily" CL (250 μg) treatments significantly delayed the onset of TNBC recurrence and suppressed tumor growth (Fig. 5b). Notably, "twice" CL treatment suppressed TNBC recurrence as efficiently as "4 daily" CL treatment, and "twice" treatment with 100 μg and 250 μg CL similarly showed a significant suppressive effect on TNBC recurrence (Fig. 5c). We therefore chose "twice" treatment with 100 μg CL as the optimal MGTRM-targeted treatment protocol for TNBC. The tumor weight of recurrent TNBC on the postoperative day 17 revealed that optimal treatment with CL significantly suppressed the growth of recurrent tumors compared with optimal treatment with PL (Fig. 5d).

Given the MGTRMs depletion by CL significantly suppressed TNBC recurrence, CL treatment would be expected to enhance the effect of chemotherapy in preventing local recurrence. Doxorubicin (Dox), the most common chemotherapy for TNBC, was therefore combined with CL using the optimal treatment protocol in the same recurrence model. Although monotherapy with Dox showed limited suppression of recurrence, which was consistent with previous reports in the treatment of TNBC patients[43], Dox + CL combination significantly ($p < 0.001$) suppressed TNBC recurrence compared with Dox monotherapy and Dox + PL treatment (Fig. 5e).

Local recurrence is often associated with distant metastasis in TNBC, and we therefore observed the lung and liver, as the most frequent metastatic sites of 4T1 tumors[44] by ex vivo bioluminescence imaging at the end of the experiment. Metastases in both tissues were reduced by all treatments, and Dox + CL combination most significantly suppressed distant metastases (Fig. 5f, g). Notably, the liver showed only background-level bioluminescence signals in the Dox + CL combination therapy group (Fig. 5g). These treatments showed no obvious adverse effects, including weight loss (Supplementary Fig. 5). These results provide the first evidence to indicate that MGTRMs could be a potential therapeutic target for TNBC.

## Discussion

Our study reported two important findings. First, MGTRMs were shown to be a major source of TAMs during early TNBC development. Second, TRMs may provide a promising target for the treatment of cancer. To the best of our knowledge, this is the first study to report that targeting MGTRMs can suppress early local recurrence after surgery and distant metastasis of TNBC. The results of this study may identify new avenues for the development of treatment for cancers for which there are currently no effective treatment targets.

Using fluorescent reporter mice and fate-mapping models, MGTRMs in the inguinal MGFPs of adult female mice were recently shown to be composed of yolk sac-derived, fetal liver-derived, and bone marrow-derived macrophage subpopulations, with distinct functions in the MGFPs[25]. These MGTRMs of different origins also express different marker proteins[31]. However, lineage-definition remains difficult due to the lack of research on MGTRMs: Most of F4/80$^+$ cells in MGFP are CD45-positeve[25] and MGTRMs would have plasticity and heterogeneity that are hallmark features of macrophages as they can rapidly adjust their functional phenotype in response to their surrounding environment[22].

Our result show that all F4/80$^+$ cells infiltrating TNBC 3 day after transplantation were CADM1-netagive, confirming that they were not myeloid-derived cells (Fig. 1f) and that half of the F4/80$^+$ cells are a FOLR2$^+$ subset. The F4/80$^+$ FOLR2$^+$ subset was reported as a favorable prognostic factor for hormone receptor positive breast cancer, and in vitro study suggested that FOLR2$^+$ MGTRMs may contribute to the activation and proliferation of CD8$^+$ T cells[31]. However, the function of FOLR2$^+$ MGTRMs in early TNBC has not yet been examined and the number of CD8$^+$ T cells was very low even at day 10 (Fig. 4e and ref. [45].), making it unlikely that FOLR2$^+$ MGTRMs contributed to T-cell-mediated anti-tumor activity in early TNBC. The existence of the other half of the FOLR2-negative MGTRMs is confirmed for the first time in this study. Although many studies of MGTRMs including us use F4/80 as a macrophage marker, other cells such as monocytes and type 2 dendritic cells are known to express F4/80[46,47]. Intensive research on MGTRMs has just begun, and it is difficult to conclude from the current limited information that the F4/80$^+$ population are entirely MGTRMs. Detailed studies on plasticity

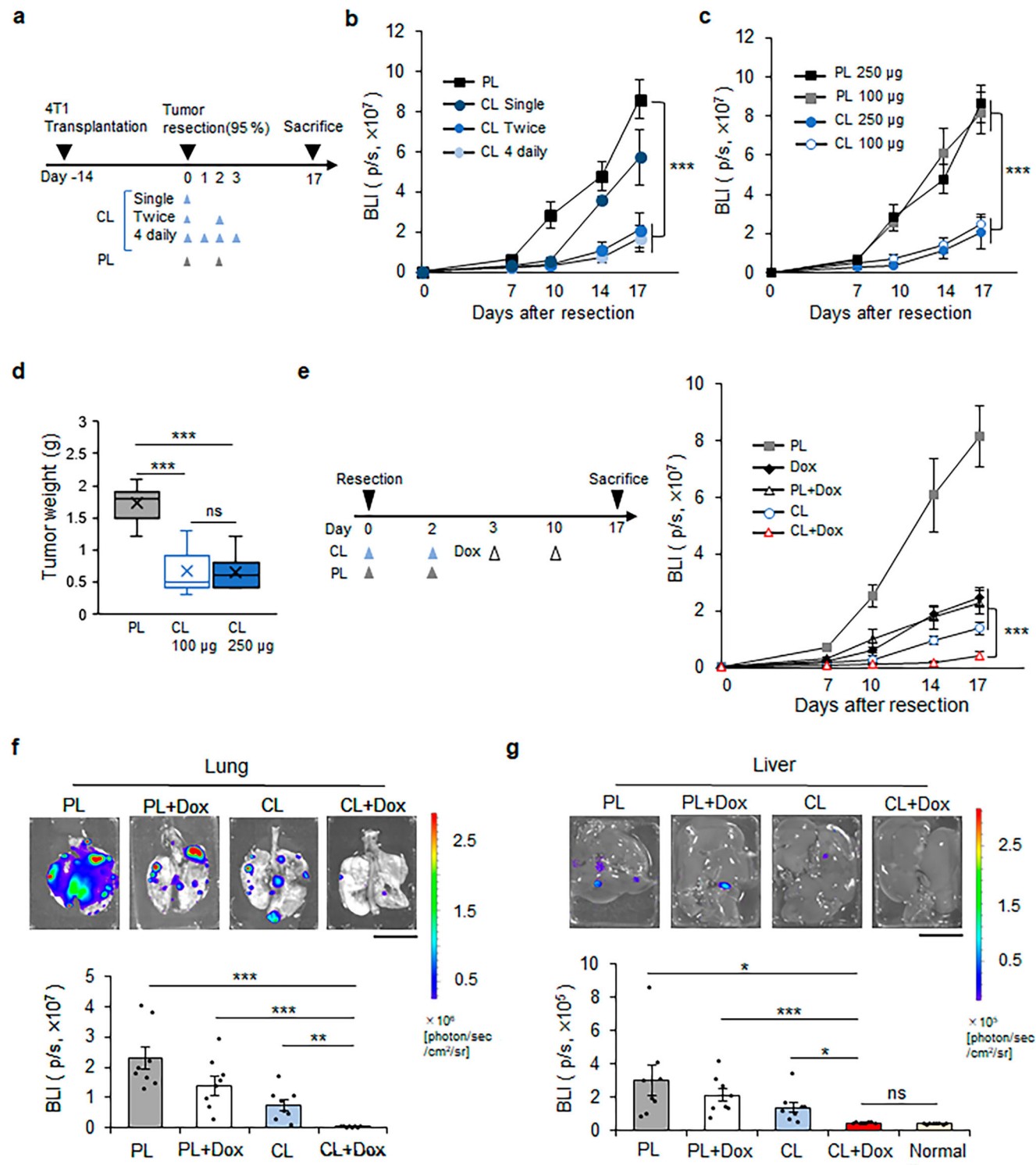

**Fig. 5 Depletion of MGTRMs suppresses the local recurrence and distal metastasis of TNBC. a** Diagram of workflow. 4T1 tumors were transplanted, resected, and treated with PL or CL administered topically to the tumor resection sites on the day of surgery (single), on the day of surgery and 2 days after surgery (twice), and for 4 consecutive days from the day of surgery (4 daily). **b** Bioluminescence signal from the 4T1 tumors left unresected was monitored after tumor resection, and treated with 250 μg PL or CL as indicated in (**a**). n = 8. **c** Bioluminescence signal of the 4T1 tumors left unresected was monitored after tumor resection and administration of CL (250 μg, 100 μg), or PL (250 μg, 100 μg) at the resection sites on the day of surgery and 2 days after surgery. n = 8. **d** Box-and-whisker plot of weight of recurrent 4T1 tumors treated "twice" with PL or CL at the endpoint (day 17). Center line: median, box limits: upper and lower quartiles, whiskers: 1.5× interquartile range. n = 7. **e** After tumor resection, PL and CL were injected at the resection sites and doxorubicin (Dox) was administered intraperitoneally at indicated days (left), and the bioluminescence signal of the 4T1 tumors left unresected was monitored (right). n = 8. **f, g** Representative images (top) and bioluminescence signals (bottom) of the lung (**f**) and liver (**g**) from the mice shown in (**e**) at the endpoint (day 17). n = 8. Scale bar = 1 cm. **b–g** ns not significant, *p < 0.05, **p < 0.01, ***p < 0.001 (t-test). **b, c, e–g** Error bars indicate s.e.m.

and heterogeneity of MGTRMs using multi-omics are expected to provide more relevant information of MGTRM subsets, leading to the promotion of research that leads to MGTRM-targeted therapy.

Single treatment with CL significantly reduced the number of MGTRMs in the MGFP for at least 10 days, and in parallel, the density of tumor vessels, number of TAMs, and distant metastases of TNBC in CL-treated MGFPs were significantly reduced compared with those in PL-treated MGFPs. These results indicate that early intratumoral infiltration of MGTRMs play a main role as TAMs in the development of the early tumor microenvironment associated with malignant progression. Epithelial-mesenchymal transition (EMT) has been shown to regulate the metastasis/invasion of cancer cells, and the tumor microenvironment plays a major role in EMT activation[48,49]. In particular, TAMs have been reported to promote EMT activation of breast cancer cells[50], suggesting that the suppression of distant metastasis by CL-treatment may be due to the reduction of the EMT in cancer cells. Further investigation of MGTRM subpopulations contributing to the development of the early tumor microenvironment may provide new insight in understanding the functional differences among MGTRM subpopulations.

The vascular density of E0771 tumors was significantly lower in CL-treated MGFPs compared with PL-treated MGFPs on day 10, while the vascular densities of 4T1 tumors were similar in both treatment group on day 10 (Fig. 4b). This finding may be explained by the difference in vascularity between 4T1 and E0771 tumors: E0771 tumors in PL-treated MGFPs had twice the vascular density of 4T1 tumors (Fig. 4b), suggesting that the number of MGTRMs recovering on day 10 was sufficient to support angiogenesis in 4T1 tumors but not in vascular-rich E0771 tumors. Furthermore, three times more TAMs infiltrated into E0771 tumors than that 4T1 tumors in PL-treated MGFPs on day 10 (Fig. 4a). The degree of TAM infiltration was previously shown to be positively correlated with the VEGF expression level, microvascular density, and vascular grade in a breast cancer model[51,52]. Indeed, E0771 tumors demonstrated a higher ability to recruit macrophages than most breast cancer cell lines, including 4T1[53]. Therefore, higher vascular density in E0771 that 4T1 tumors may thus have been due to the abundant TAM infiltration.

Recurrence tends to occur early after surgery, followed by rapid disease progression and distant metastases, leading to poor outcomes of TNBC patients[43]. Systemic chemotherapy is often used to reduce these risks[43]. However, the emergence of chemoresistance is more common in TNBC compared with non-TNBC patients[54], and the prevention of postoperative local recurrence remains one of the main treatment challenges[43,55]. TNBC shows considerable overlap with *BRCA1*-mutated tumors, and 75%–85% of women with *BRCA1* germline mutation-associated breast cancer have the TNBC subtype[56]. Therefore, patients with this type of tumor are at potentially high risk of ipsilateral or contralateral recurrence, and risk-reduction mastectomy is strongly considered in patients with *BRCA1* mutation[57]. However, an investigation of long-term patient-reported outcomes after surgical treatment of early-stage breast cancer showed equivalent patient satisfaction with their breasts and physical well-being, but better psychosocial and sexual well-being, in patients who underwent breast-conserving surgery with subsequent radiation therapy compared with patients who underwent mastectomy and reconstruction[58]. Using a 4T1 local recurrence model, we successfully showed that the depletion of MGTRMs by CL treatment at tumor resection significantly reduced local recurrence and distance metastases, and enhanced the chemotherapeutic outcome (Fig. 5). These results provide a potential treatment option for TNBC patients using TRM-targeted therapy, which may facilitate patient choice in term of breast conservation.

Previous studies have reported that TRMs are involved in tumor immunosuppression in lung cancer models via the expansion of regulatory T cell (Treg)[20]. However, they have analyzed lung cancers 30 days after the intravenous administration of cancer cells. Foxp3$^+$ Treg cell infiltration into 4T1 and E0771 tumors is known to be extremely low during early tumor development (around 20 days after transplantation) and to only reach maximal levels at later stages (>30 days after transplantation)[45]. The percentage of functional CD11a$^{high}$CD8$^+$ T cells among CD8$^+$ T cells peaks around 14 days after 4T1 cell transplantation[59]. These facts indicate that, if MGTRMs are involved in tumor immunosuppression, it would likely be at later stages (>20 days after transplantation). We therefore did not investigate whether MGTRM-targeted therapy enhances the effects of immune checkpoint inhibitors, despite their being a current focus of attention, because these are expected to be effective for tumors with high numbers of tumor-infiltrating lymphocytes[60].

Although the effects of CL on dendritic cells have been reported[40], there are very few dendritic cells in MGFP[35] (Supplementary Fig. 3d), and even if they were inhibited, their effect on early TNBC would be limited. The low infiltration of CD8$^+$ T cells in early TNBC[45] and the absence of changes in CD8$^+$ T cells within TNBC transplanted into PL/CL-treated mammary glands make it unlikely that dendritic cell-mediated acquired immunity would affect the growth of early TNBC. Therefore, the involvement of mammary tissue-resident dendritic cells in our TNBC model is likely to be small.

Although our study was restricted to TNBC, targeting MGTRMs is expected to be an effective approach for preventing early recurrence after surgery of other breast cancer subtypes. There is, however, currently no topical clinical treatment option for breast cancer. Because systemic TRM-targeted therapy is not suitable because of its side effects caused by widely targeting phagocytic cells, advances in clinical research on topical administration are desired. Postoperative administration of TRM-targeted therapeutic agents to the surgical sites is expected to provide a well-controlled treatment option, suitable for use in combination with other therapies, such as radiation therapy. Tissue-specific gene expression has been confirmed for TRM[22]. Identification of markers specific to each TRM population may facilitate systemic treatment strategies for particular cancer by targeting specific TRM.

## Methods

**Mice**. C57BL/6J (B6), B6(Cg)-Tyrc-2J/J (B6 albino), and BALB/c mice were obtained from Charles River Laboratory Japan (Yokohama, Japan). GFP-Tg mice were produced by introducing a fragment of CAAG-GFP into fertilized eggs of B6 mice. All mice had access to food and water *ad libitum* and were housed in the animal facilities at the Tokyo Institute of Technology and Kyoto University. Age-matched females, 8–10 weeks old, were used in all experiments. All experiments using mice were approved by the Animal Experiment Committees of the Tokyo Institute of Technology and Kyoto University and carried out in accordance with relevant national and international guidelines. The experiments were stopped when the tumor diameter reached 2 cm.

**Cells and culture conditions**. E0771 and 4T1 murine breast cancer cell lines were purchased from CH3 BioSystems (Buffalo, NY) and ATCC (Manassas, VA), respectively. E0771/mKO2-luc2 and 4T1/Fluc cells were established as described previously[61]. Cells were maintained with 5% fetal bovine serum (FBS)-Dulbecco's Modified Eagle's Medium (Nacalai Tesque, Kyoto, Japan) supplemented with penicillin (100 units/ml) and streptomycin (100 mg/ml), cultured in a 5% $CO_2$ incubator at 37 °C, and regularly checked for mycoplasma contamination using a mycoplasma detection kit (Lonza, Basel, Switzerland). All cell lines were independently stored and recovered from the original stock for each experiment.

**Orthotopic breast cancer model**. Cell suspensions of E0771/mKO2-luc2 ($5.0 \times 10^5$ cells) and 4T1/Fluc ($3.0 \times 10^5$ cells) in PBS were mixed with an equal

volume of Geltrex® (Invitrogen, Waltham, MA) and injected into the fourth MGFP of 8–10-week-old B6 albino and BALB/c female mice.

**Immunohistochemistry.** For preparing tumor sections, first, MGFP containing a tumor was removed from mice at the indicated day after orthotopic transplantation of TNBC cells. The MGFP was placed in a container, immersed in 100 µg/ml D-luciferin solution (Promega, Madison, WI), and the tumor was removed from the MGFP with reference to bioluminescence images obtained with IVIS®-Spectrum (Perkin Elmer, Waltham, MA). Tumor tissues were immediately frozen in optimum cutting temperature compound (Sakura Finetek Japan, Tokyo, Japan), cryosectioned (10 µm) using a Leica CM1850 cryostat (Leica Biosystems, Nussloch, Germany), and fixed in 4%-Paraformaldehyde PBS (Nacalai Tesque) for 10 min at room temperature. After blocking with blocking buffer (3% bovine serum albumin in PBS) for 30 min at room temperature, tumor sections were incubated with primary antibodies diluted in blocking buffer overnight at 4 °C, and with fluorochrome-conjugated secondary antibodies for 1 h at room temperature in the dark. Nuclei were stained with bisbenzimide H33342 fluorochrome trihydrochloride (Nacalai Tesque), and sections on slides were mounted with Fluoromount™ (Diagnostic BioSystems, Pleasanton, CA). Images were obtained with a BZ-X710 microscope (Keyence, Osaka, Japan), and Image J[62] was used to quantify fluorescence-positive cells. After adjusting the brightness and contrast, the image was converted to 8-bit grayscale. Thresholds were set for GFP-, F4/80-, α-SMA-, CD206-, FOLR2-, CD3-, and CD11c-positive cell measurements, and connected cells were separated into single cells using the watershed-based cell segmentation[63] before counting. For CD31+ measurements, no watershed-based cell segmentation was used, and the positive areas were quantified. The percentages of F4/80- and α-SMA-positive cells was calculated by dividing the number of F4/80- and α-SMA-positive cells by the total number of GFP-positive cells. The percentage of FOLR2- and CD206-positive cells was similarly calculated by dividing the number of FOLR2- and CD206-positive cells by the total number of F4/80 positive-MGTRMs.

The following antibodies were used: α-F4/80 (clone BM8, 1:50; BioLegend, San Diego, CA), α-GFP (polyclonal, 1:250; Abcam, Cambridge, UK), α-a-SMA (clone: 1A4, 1:100; Sigma-Aldrich, St. Louis, MO), α-CD31 (clone MEC 13.3, 1:100; BD Biosciences, Franklin Lakes, NJ), α-FOLR2 (polyclonal, 1:200; Novus biological, Centennial, CO), α-CD206 (polyclonal, 1:1000; Abcam), α-CADM1 (clone: 3E1, 1:100; MBL, Tokyo, Japan), α-CD3 (clone: 17A2, 1:150; R & D systems, Minneapolis, MN), α-CD11c (clone: N418, 1:100; Bio-Rad Laboratories, Hercules, CA), anti-mouse IgG-Alexa fluor 488, anti-rat IgG-Alexa fluor 488, anti-chicken IgY-Alexa Fluor 647, anti-rabbit IgG-Alexa Fluor 546 (1:500; Thermo Fisher Scientific, Waltham, MA), and anti-hamster IgG- Alexa fluor 488 (1:500; Abcam).

For preparing MGFP sections, the fourth pair of MGFP was removed, fixed in 4%-Paraformaldehyde Phosphate Buffer, paraffin-embedded, and cut into 4 µm-thick sections. Ly-6G immunostaining of the sections was conducted by Morpho Technology, Inc (Sapporo, Japan). Briefly, pretreated sections were incubated with biotinylated anti-rat IgG antibody (Vector, Newark, CA) at room temperature for 30 min, followed by VECTASTAIN Elite ABC reagent at room temperature for 30 min, and Tris-buffered saline 3,3′-diaminobenzidine for 5 min at room temperature for color development. The sections were counterstained with hematoxylin. After staining, the sections were observed under a BZ-X710 microscope, and the numbers of Ly-6G positive cells was counted.

**Proliferation assay for TNBC cells co-cultured with MGFP and MGTRMs.** TNBC cells (1.0 × 10⁴/500 µl serum-free medium) were seeded in the bottom chamber, and MGFP was cultured with 100 µl serum-free medium (covering 80% of the surface) in the upper chamber of a 6.5 mm transwell® chamber with a 0.4 µm pore membrane insert (Cat. No. 3470, Corning, NY). PL or 250 µg of CL (Katayama Chemical, Osaka, Japan) were added to the upper chamber at the start of co-culture with MGFP. For co-culture with MGTRMs, MGTRMs (1.0 × 10⁵ cells/100 µl serum-free medium) were seeded in a 24-well culture dish with 1 × 10⁴ TNBC cells. After 3 days of incubation, cells were lysed with Passive Lysis Buffer, and luciferase activity was measured with a Luciferase Assay Kit (Cat. No. E1501, Promega, Madison, WI) according to the manufacturer's instructions using the GL-210A luminometer (Microtec, Chiba, Japan).

**CL treatment to TNBC cells.** TNBC cells (1 × 10⁴/well) were seeded in 24-well plates, and CL (250 µg) were added to each well. After 3 days of incubation, luciferase activity in TNBC cells was measured using a Luciferase Assay Kit with a GL-210A luminometer.

**Preparation of single-cell suspension from the MGFP.** The fourth MGFP was removed from 8–10-week-old female mice, minced, and digested in Hank's Balanced Salt Solution (Nacalai Tesque) containing 500 µg Liberase DH (Roche, Basal, Switzerland) and 0.16 µl DNase I (Clontech Laboratories, Mountain View, CA) at 37 °C for 2 h. Then, a single-cell suspension was obtained by sequentially passing samples through a silk-based membrane with a pore size of 77 µm (Sansyo, Tokyo, Japan) and strainers with a pore size of 40 µm (Greiner Bio-one, Kremsmünster, Austria). After lysing red blood cells with PharmLyse solution (BD Biosciences) for 2 min at room temperature, cells were resuspended in FACS buffer (5% FBS and 2 mM EDTA in PBS).

**Preparation of single-cell suspension from tumors.** 4T1 and E0771 tumors were removed from female mice, minced, and digested in 2% FBS/RPMI-1640 (Nacalai Tesque) containing 500 µg Liberase DH and 0.16 µl DNase I at 37 °C for 1 h. A single-cell suspension was then obtained by sequentially passing samples through strainers with pore sizes of 100 and 40 µm. After lysis of red blood cells with PharmLyse solution for 2 min at room temperature, the cells were resuspended in FACS buffer (5% FBS and 2 mM EDTA in PBS).

**Flow cytometry and cell sorting of MGTRMs.** Tumor singlets and MGFP suspensions in FACS buffer were treated with α-CD16/32 (clone 93, 1:200; BioLegend) at 4 °C for 20 min to block Fc receptors, followed by incubation with fluorescent-labeled antibodies at 4 °C in the dark for 25 min in FACS buffer. The following antibodies were used: α-CD11b (clone M1/70, 1:100; BioLegend), α-F4/80 (clone CI:A3-1, 1:20; Bio-Rad Laboratories) for MGTRMs, α-CD11c (clone N418, 1:100; eBioscience, San diego, CA), α-MHC-II (clone M5/114.15.2, 1:100, BD Bioscience) for dendritic cells, α-CD11b (1:100), α-Ly-6c (clone HK1.4, 1:200; BioLegend) for tumor-infiltrating monocytes, α-CD8a (clone 53-6.7, 1:100; eBioscience) for T-cells, and α-NK1.1 (clone PK136, 1:100; eBioscience) for natural killer cells. The antibody-labeled cells were sorted and analyzed using an SH800Z fluorescence activated cell sorter and EC800 flow cytometry analyzer (Sony, Tokyo, Japan), respectively.

**Treatment of MGTRMs with CL in vivo.** For in vivo experiments, 100 or 250 µg of CL were topically administered to the fourth MGFP of mice anesthetized with isoflurane (Wako Pure Chemical, Osaka, Japan). CL were injected 1 day before orthotopic transplantation of TNBC cells when examining their effect on tumor growth, and 2 and 4 days after tumor resection to evaluate their effect on recurrence, unless indicated otherwise. PL was administered as a control in each in vivo experiment.

**In vivo bioluminescence imaging.** Mice with E0771/mKO2-luc2 and 4T1/Fluc tumors were intraperitoneally injected with 100 µl of 100 µg/ml D-luciferin solution (Promega) and sequentially imaged every 3 min for 21 min using an in vivo photon-counting device IVIS®-spectrum, and images with the maximum photon counts were used for analysis. The following conditions were used for image acquisition: exposure time = 1 min, binning = medium: 8, field of view = 22.5 × 22.5 cm, and f/stop = 1. The minimum and maximum photon/second/cm²/steradian (p/s/cm²/sr) for each image are indicated in each figure by a rainbow bar scale.

**Photoacoustic imaging of 4T1 tumors.** Mice euthanized with isoflurane were imaged using a Photoacoustic 3D imaging system (Luxonus, Kanagawa, Japan). Imaging and data acquisition were performed as described previously[64]. Briefly, images were acquired using an alternating irradiation mode at wavelengths of 797 nm and 756 nm. The water temperature was set at 37 °C, and the sound speed for reconstitution was set automatically. The photoacoustic images were reconstructed using Universal Back Projection[65].

**Mouse TNBC recurrence model.** When the diameter of TNBC reached 1 cm, 95% of the tumor was removed as previously described[42]. Subsequent tumor recurrence and growth were assessed by in vivo bioluminescence imaging.

**Dox treatment.** Dox (Wako Pure Chemical) was administered intraperitoneally once a week from the day 3 after tumor resection at the previously reported dose of 5 mg/kg[66].

**Ex vivo bioluminescence imaging of the lung and liver.** Mice anesthetized with isoflurane were intraperitoneally injected with 100 µl of 100 µg/ml D-luciferin solution, and the lungs and livers were removed 15 min later. The excised lungs and livers were placed in a container, further immersed in 100 µg/ml D-luciferin solution, and sequentially imaged every 2 min for six times with an IVIS®-Spectrum. Images with the maximum photon counts were used for analysis. The following conditions were used for image acquisition: exposure time = 1 min, binning = small, field of view = 6.5 cm × 6.5 cm, and f/stop = 1. Imaging data were quantitatively analyzed using Living Image 4.3 software. The minimum and maximum photons/s/cm²/sr of each image are indicated in each Figure by a rainbow bar scale.

**Statistics and reproducibility.** The statistical significance between values was determined by the two-sided unpaired Student's t test. All data were expressed as the mean ± standard error of the mean. Probability values (p values) of 0.05 or less were considered significant. All experiments were independently repeated two or more times to confirm reproducibility.

**Reporting summary**. Further information on research design is available in the Nature Portfolio Reporting Summary linked to this article.

## Data availability

The source data underlying Figs. 1a–c, e, 2, 3, 4a, b, d–g, 5b–g, Supplementary Figs. S1, S3c, d, S5 are provided as Supplementary Data. All other data generated during this study are available from the corresponding author on reasonable request.

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

## Acknowledgements

The authors thank Dr. Yasufumi Asao (luxonus Inc. https://www.luxonus.jp/) for technical support of photoacoustic imaging, and the Open Research Facilities for Life Science and Technology, Tokyo Institute of Technology for supporting immunohistochemical analysis. This study was supported by the Princess Takamatsu Cancer Research Fund (S. K.-K.), the Uehara Memorial Foundation (S. K.-K.), the NOVARTIS Foundation of Japan (S. K.-K), and MEXT Quantum Leap Flagship Program (MEXT QLEAP) Grant Number JPMXS0120330644 (S. K.-K.). We thank Melissa Crawford, PhD, from Edanz (https://jp.edanz.com/ac) for editing a draft of this paper.

## Author contributions

R.H., K.O., M.S., M.Sa., and S.W. performed the experiments, and collected and analyzed the data; R.H., T.K., and S.K.-K. contributed to designing the experiments, interpreting the data; H.W. and G.K. constructed and maintained the GFP-Tg mice; R.H. and S.K.-K. wrote the paper. All authors contributed to the interpretation of the results and read and approved the paper.

## Competing interests

The authors declare no competing interests.
