## [Peer Review File · Communications Biology]

Reviewers' comments:

Reviewer #1 (Remarks to the Author):

In this study, authors investigate the role of mammary gland tissue-resident macrophages (MGTRM) in the development of TNBC breast tumors and recurrence after surgery and chemotherapy. To this end they combine immunofluorescence of macrophages within neoplastic mammary glands, various tumor models and the use of clodronate liposome for MGTRM depletion in vivo and assessment of tumor growth, recurrence and metastasis.

In the introduction, authors make several claims that are not correct

First, authors say that TRM are all from embryonic origin. This is not correct, TRM not necessarily originate from embryonic precursors. They can also derive from monocytes in lungs (Chakarov et al Science 2019), gut (Bain et al, Nature Immunology 2018) or heart (Bajpai et al, Nature Medicine 2018)

Secondly, authors say that MGTRM function has not been studied before in breast cancer. This is untrue since the following studies have investigated the role of TRM in mammary cancers

Linde, Nat Com 2018 (PMID: 29295986)

Franklin, Science 2014 (PMID: 24812208)

Nalio-Ramos Cell 2022 (PMID: 35325594)

In addition, several concerns limit the robustness of the conclusions

Fig1b

To be exact, quantification of immune infiltrate should be performed by flow cytometry and not by immunohistochemistry.

Moreover, current knowledge enables to know whether macrophages are TRM or not using the following markers: CD11c, MHC-II, CD11b, F4/80, CD64, CD206 and FOLR2. Authors should provide a more complete panel of markers and perform flow cytometry to characterize the macrophage infiltrates

Also using flow cytometry authors should tell us what are the "other" cells. Are they CD45+ hematopoietic cells? Are they CD11c+CD26+ dendritic cells?

Fig2A

Clodronate not only depletes macrophages but any phagocytic cells.

Authors should provide a control with clodronate treatment only to see if CL does not lead to tumor cell death directly

Supp Fig 2a: CL depletes also dendritic cells which could play a major role in tumor development.

Therefore, authors should also look at depletion of dendritic cells (CD11c+MHCII+CD26+ but negative for CD64)

Fig 4e

Again, authors should be more specific when addressing the phenotype of TRM versus recruited monocyte-derived TAMs

Various markers including CD11c, MHC-II, CD11b, F4/80, CD64, CD206 and FOLR2 have been used to distinguish TAMs population

Authors should use them to assess the phenotype of TAMs at early time points (day1) in order to claim undoubtedly that macrophages are TRM

This study (Linde et al. – Nature Communication 2018 - Macrophages orchestrate breast cancer early dissemination) has already shown that TRM permit early dissemination and therefore metastasis. How is the current study different?

Minor comment

In the introduction authors include hematopoietic immune cells in "Tumor-infiltrating stromal cells". This is misleading because stromal cells are non-hematopoietic cells like endothelial cells and fibroblasts e.g. Immune cells are not stromal cells.

Reviewer #2 (Remarks to the Author):

In the manuscript titled "Tissue-resident macrophages are major tumor-associated macrophage resources contributing to the early development TNBC development, recurrence, and metastases," the authors aim to define the pro-tumorigenic role of resident macrophages by depleting this population of cells and measuring distinct aspects of tumorigenesis including proliferation and angiogenesis. While the authors make a compelling argument, there are some aspects that need further analysis before the manuscript is considered ready for publication.

Major comments:

While clodronate liposomes are widely used for the depletion of macrophages, the authors need to show clearly that these cells are absent in the mammary gland. The authors show immunohistochemistry in 3A only by using F4/80+ immunoreactivity. There should be flow cytometry data from the mammary gland and from tumor-draining lymph nodes for markers of circulating and tissue-resident macrophages to determine if the depletion or level of depletion took place. (There is flow cytometry data in the supplemental portraying the gating strategy for the acquisition of resident macrophages; the depletion data is missing or absent).

Several studies, including some cited by the authors, have studied the role of tissue-resident macrophages in cancers, including TNBC. The authors in the introduction claim that "Although various studies have investigated TAM-specific targeting, a successful approach has not been found as all the approaches to date have limitations." According to figure 3A the treatment with CL only reduces the tissue-resident macrophages by 50%. So it is hard to understand what the study is contributing, given that is plagued by the same limitations.

For in vivo experiments, the authors use topical administration of CL. The manufacturer's instructions indicate IV injection. A quick review of the literature suggests SC injections as well. Perhaps due to the route of administration, the only observed 50% depletion, which then limits the conclusions made in the manuscript.

Minor comments:

The fluorescent imaging figures should also include a merge figure that includes the nuclei and the staining (figures 1a-1c; 4a-4b).

The data in figure 1 should also be expressed in the context of tumor size.

In figure 2, what is the percentage of death of MGRM after CL treatment relative to cancer cell growth?

The presentation of data from 4T1 and E0071 should be consistent throughout the manuscript, not choose one or the other for supplemental.

The cytokine data in figure 2 is an afterthought; there is no follow-up to mechanisms. At least this should be explained in the discussion in the context of defined lineages of Balb/c and C57Bl6.

Reviewer #3 (Remarks to the Author):

This study aimed to investigate the role of resident macrophages in early TNBC development, recurrence, and metastases. The data are convincing; however, the paper still need to be revised.

- In Figure 2, I don't understand why the TNBC cell line was cultured separated from the macrophage

since they are in close contacts in the tumor. This need to be clarified. Moreover, the authors used a TNBC cell line expressing the luciferase to evaluate its proliferation. To be able to use the luciferase activity as a readout for proliferation in vitro, the authors must show that the MGFP secretome, the CL and PL has no effect on the expression of the luciferase RNA in the TNBC cell line. A standard proliferation assay by counting the cell would be more appropriate.

- I feel that assessment of the cytokines secretion in this study is dispensable and does not bring any clues.

Response to Reviewers

Reviewer #1 (Remarks to the Author):

In this study, authors investigate the role of mammary gland tissue-resident macrophages (MGTRM) in the development of TNBC breast tumors and recurrence after surgery and chemotherapy. To this end they combine immunofluorescence of macrophages within neoplastic mammary glands, various tumor models and the use of clodronate liposome for MGTRM depletion in vivo and assessment of tumor growth, recurrence and metastasis.

Thank you for pointing out so many important issues and suggesting crucial experiments. They greatly improved our manuscript. The revised major words and sentences are underlined in the text of the revised manuscript.

1) In the introduction, authors make several claims that are not correct

First, authors say that TRM are all from embryonic origin. This is not correct, TRM not necessarily originate from embryonic precursors. They can also derive from monocytes in lungs (Chakarov et al Science 2019), gut (Bain et al, Nature Immunology 2018) or heart (Bajpai et al, Nature Medecine 2018)

Thank you for pointing out the misleading expression. We have changed the wording in the revised manuscript as follows: “TRMs have several origins: yolk sac macrophages, fetal liver monocytes, or adult bone-marrow monocytes, and locally self-replicate independently of hematopoietic stem cells in adult tissues,” (lines 42 – 43)

2) Secondly, authors say that MGTRM function has not been studied before in breast cancer. This is untrue since the following studies have investigated the role of TRM in mammary cancers

Linde, Nat Com 2018 (PMID: 29295986)

Franklin, Science 2014 (PMID: 24812208)

Nalio-Ramos Cell 2022 (PMID: 35325594)

Thank you again for pointing out the misleading expression. I agree with the reviewer that some studies have investigated the role of MGTRMs in mammary cancers and these studies, including the studies provided by the reviewer, extensively characterize the MGTRMs infiltrated to breast cancers. However, none of them have investigated the therapeutic efficacy of targeting MGTRMs. A novel finding of our study is that MGTRMs play a major role in early growth and recurrence of TNBC, clearly showing that MGTRMs are a promising therapeutic target. To avoid misunderstandings, the text of the introduction has been revised

as follows, citing references provided by the reviewer: “Although some studies have investigated the involvement of MGTRMs in breast cancers (references), the significance of MGTRMs as a therapeutic target in breast cancer has remained unexplored.” (lines 50 – 52)

3) Fig1b

To be exact, quantification of immune infiltrate should be performed by flow cytometry and not by immunohistochemistry.

Moreover, current knowledge enables to know whether macrophages are TRM or not using the following markers: CD11c, MHC-II, CD11b, F4/80, CD64, CD206 and FOLR2. Authors should provide a more complete panel of markers and perform flow cytometry to characterize the macrophage infiltrates

Thank you for suggesting flow cytometry analysis. We agree with the reviewers that quantitative analysis with flow cytometry provides more accurate information. However, tumors used in experiments up to 3 days post-implantation were too small to be detected by the naked eye. Tumors were therefore collected for immunohistochemical analysis while *ex vivo* bioluminescence imaging was used to confirm tumor location. Unfortunately, it is therefore difficult to analyze tumor cells by flow cytometry. Instead, we analyzed the expression of markers in tumor sections by immunohistochemistry. The results are shown in Fig. 1 d, 1e, and 1f in the revised manuscript. We have added some sentences in the Results and Discussion sections (lines 81 – 87, and 211 – 221). We have also added the tumor sizes used in these analyses to the legend of Figure 1 and supplementary Fig. 1, and added the method for resecting day-1 to day-3 tumors from mice to the Method section (lines 325 -329).

Thanks to you for suggesting to analyze FOLR2 and CD206 expression in early TNBC, we found that approximately half of MGTRMs was FOLR2/CD206⁺ subpopulation of MGTRMs. Although we have no information about the other half of the MGTRMs, we were able to clearly demonstrate that a FOLR2-negative MGTRM subpopulation exists in MGFP.

4) Also using flow cytometry authors should tell us what are the “other” cells. Are they CD45⁺ hematopoietic cells? Are they CD11c⁺CD26⁺ dendritic cells?

Thank you for suggesting analysis of “other” cell populations by flow cytometry. Fig. 1b shows the data from immunohistochemical analysis of tumor 3 days after transplantation. For the same reason as our response in 3), unfortunately flow cytometry analysis is almost impossible. Instead, we analyzed CD11c⁺ dendritic and CD3⁺ T cells known to present in mammary tissue (dendritic cells, PMID: 29884705; T cells, PMID: 32402923) by immunohistochemistry and found that the number of these cells infiltrating TNBC tumor 3 days after transplantation was very low compared to MGTRMs (Fig A). Although the results

shown in Fig. A could not clarify all the “other” cells, our conclusion that the macrophage population is the predominant cell population infiltrating the tumor 3 days after transplantation remains the same. The results are shown in Supplementary Fig.1. We have added sentences in the Result and Discussion sections (lines 77 – 79 and 278 -284).

Because most of the F4/80-positive cells in mammary tissue are also CD45⁺ (Fig B, also see PMID: 30655530), CD45 is not suitable as a hematopoietic cell marker here.

[Fig A] Left: Sections of 4T1 tumors in BALB/c mice 3 days after transplantation. Representative fluorescence images of CD11c⁺ and CD3⁺ cells. Scale bar = 50 μm. **Right:** Average number of F4/80⁺, CD3⁺ cells and CD11c⁺ cells in a field. n = 15

[Fig B] Flow cytometry analysis of CD45 and F4/80 expression in live cells from BALB/c MGFP. The dotted circle indicates CD45⁺F4/80⁺ MGTRMs. Very few CD45⁺F4/80⁺ cell population was detected in MGFP.

5) Fig2A

Clodronate not only depletes macrophages but any phagocytic cells.

Authors should provide a control with clodronate treatment only to see if CL does not lead to tumor cell death directly

Reviewer #3 made a similar comment, but the results showing that CL did not affect TNBC cell viability were presented in Supplementary Fig. 2c of the original manuscript. We realized that the results were important rather than supplementary, so we have moved the result to Fig. 2c.

6) Supp Fig 2a: CL depletes also dendritic cells which could play a major role in tumor development. Therefore, authors should also look at depletion of dendritic cells (CD11c⁺MHCII⁺CD26⁺ but negative for CD64)

Thank you for pointing out an important issue. We analyze dendritic cells in mammary fat pad by flow cytometry. The results indicated that CD11c⁺ MHCII⁺ cell population in MGFP was ~0.59%, much less than MGTRMs (~13.1%) [Supplementary Fig. 3d and Fig. 3a in the revised manuscript]. Furthermore, in the early tumors, the abundance of CD8⁺ T cell was also very low [Fig. 4e in the revised manuscript]. Therefore, we believe that the contribution of dendritic cells to TNBC growth suppression is limited. As we consider the result of mammary tissue-resident dendritic cells to be important information, we have added the data to Supplementary Fig. 3d and revised the text in the Results and Discussion sections of the revised manuscript (lines 114 -119 and 278 -284).

7) Fig 4e

Again, authors should be more specific when addressing the phenotype of TRM versus recruited monocyte-derived TAMs

Various markers including CD11c, MHC-II, CD11b, F4/80, CD64, CD206 and FOLR2 have been used to distinguish TAMs population

Authors should use them to assess the phenotype of TAMs at early time points (day1) in order to claim undoubtedly that macrophages are TRM

Fig. 4e [Fig. 4g in the revised manuscript] shows the results of the normal mammary tissue of tumor-free mice to know the time required for MGTRMs recovery after CL treatment. Therefore, TAMs were not included in this population. To avoid misleading, we have revised the corresponding text as follows: “To verify this, the number of F4/80⁺ cells in MGFPs treated with CL was examined by flow cytometry 1 day and 11 days after the topical administration of CL to tumor-free mice” (lines 156 - 158).

8) This study (Linde et al. – Nature Communication 2018 - Macrophages orchestrate breast cancer early dissemination) has already shown that TRM permit early dissemination and therefore metastasis. How is the current study different?

Thank you for the comment on important issues. I agree that Linde et al. clearly showed that a causal role for macrophages in early dissemination and elucidated the molecular mechanism to drive early dissemination of HER2⁺ breast cancer. The model they used was the MMTV-HER2 Tg mouse model, and MGTRM attraction by early cancer cells from the stroma into the epithelial layer of lesions (defined as mammary intra-epithelial neoplasia) in mice depends on HER2-NF-κB-mediated induction of CCL2.

Linde et al. more extensively investigated the molecular mechanism of dissemination from early cancer lesion using mouse model more relevant to clinical breast cancer. However, as it is difficult to determine exactly when and where cancer cells develop, it is also difficult to

determine when and where cancer cells initiate interaction with MGTRMs and TAMs. In other words, they cannot be completely ruled out the possibility that the points they were observing were influenced by nearby tumors: The macrophages they observed could be tissue-resident macrophages or TAMs that migrated from nearby tumors. On the other hand, in our experiment, although it is more artificial model, it is clear that cancer cells come into contact with MGTRMs for the first time at the time of transplantation. This system allowed us to observe the contribution of (TAM-free) MGTRMs on TNBC proliferation. As described above (for comment 2), a novel finding of our study is that MGTRMs play a major role in early growth and recurrence of TNBC, clearly indicating that MGTRMs are a promising therapeutic target.

Minor comment

9) In the introduction authors include hematopoietic immune cells in “Tumor-infiltrating stromal cells”. This is misleading because stromal cells are non-hematopoietic cells like endothelial cells and fibroblasts e.g. Immune cells are not stromal cells.

Thank you for the advice. Immune cells are also included in “stroma cells” in a broad sense. For example, the abstracts of the following reviews describe stromal cells in such a broad sense. We would like to use “stromal cells” in a broad sense.

Immunity, 2021 May 11;54(5):885-902. doi: 10.1016/j.immuni.2021.03.022.

Nat Rev Cancer, 2019 Aug;19(8):454-464. doi: 10.1038/s41568-019-0168-y.

Reviewer #2 (Remarks to the Author):

In the manuscript titled “Tissue-resident macrophages are major tumor-associated macrophage resources contributing to the early development TNBC development, recurrence, and metastases,” the authors aim to define the pro-tumorigenic role of resident macrophages by depleting this population of cells and measuring distinct aspects of tumorigenesis including proliferation and angiogenesis. While the authors make a compelling argument, there are some aspects that need further analysis before the manuscript is considered ready for publication.

We appreciate your valuable advice and efforts to improve our manuscript. The revised major words and sentences are underlined in the text of the revised manuscript.

Major comments:

1) While clodronate liposomes are widely used for the depletion of macrophages, the authors need to show clearly that these cells are absent in the mammary gland. The authors show immunohistochemistry in 3A only by using F4/80+ immunoreactivity. There should be flow

cytometry data from the mammary gland and from tumor-draining lymph nodes for markers of circulating and tissue-resident macrophages to determine if the depletion or level of depletion took place. (There is flow cytometry data in the supplemental portraying the gating strategy for the acquisition of resident macrophages; the depletion data is missing or absent).

Thank you for suggesting flow cytometry analysis of MGTRMs in MGFP after CL treatment. We agree with the reviewers that flow cytometry analysis provides more accurate information. The reason for using immunohistochemical analysis was that we were unable to successfully extract MGTRMs from MGFP and did not obtain adequate cell numbers for flow cytometry analysis. As a result of examining various extraction conditions, we were able to establish a method for obtaining a sufficient number of MGTRMs for flow cytometry analysis, so we performed analysis using flow cytometry. Flow cytometry result shows more significant reduction (~86%) of MGTRMs. We have replaced the immunohistochemical analysis results in Fig. 3a with the flow cytometry plots in Supplementary Fig. 3a and revised the corresponding text in the revised manuscript (lines 106 - 109). We were unable to find the reason why the difference in analysis methods cause such a large difference. Staining with the ABC method requires knowledge of pathology in order to accurately count positive cells, and currently no software for counting positive cells is available. It is possible that our initial method of counting positive cells may not be appropriate, as our knowledge of pathology may not be enough.

Fig. 3a shows the results of mammary tissue from tumor-free mice. Therefore, we did not analyze tumor-draining lymph nodes.

2) Several studies, including some cited by the authors, have studied the role of tissue-resident macrophages in cancers, including TNBC. The authors in the introduction claim that “Although various studies have investigated TAM-specific targeting, a successful approach has not been found as all the approaches to date have limitations.” According to figure 3A the treatment with CL only reduces the tissue-resident macrophages by 50%. So it is hard to understand what the study is contributing, given that is plagued by the same limitations. For in vivo experiments, the authors use topical administration of CL. The manufacturer’s instructions indicate IV injection. A quick review of the literature suggests SC injections as well. Perhaps due to the route of administration, the only observed 50% depletion, which then limits the conclusions made in the manuscript.

As responded to your comment 1), we found that CL treatment reduced MGTRMs by about 86%. This reduction is more consistent with the results of the numbers of TAMs in tumors on day 4 and day 10 shown in Fig. 4a: TAMs were significantly reduced (more than 10 times) by CL treatment compared to PL treatment. Topical administration of CL caused a marked

decrease of TAMs in TNBC, without obvious side effects such as weight loss (Supplementary Fig. 5). Therefore, we believe that the route of administration was appropriate in our experiments.

Minor comments:

3) The fluorescent imaging figures should also include a merge figure that includes the nuclei and the staining (figures 1a-1c; 4a-4b).

Thank you for the advice. Figures 1a-1c and 4a-4b have been modified accordingly.

4) The data in figure 1 should also be expressed in the context of tumor size.

Thank you for your advice. We added the tumor size information in Figure 1 and Supplementary Figure 1 legends.

5) In figure 2, what is the percentage of death of MGRTM after CL treatment relative to cancer cell growth?

In the experiment for Fig. 2a, unfortunately, it is impossible to count the number of MGTRMs in MGFP before and after CL treatment in each well. Therefore, it is difficult to assess the viability of MGTRMs in this assay. Since we determined, however, the dose of CL for each well based on *in vivo* experiment, we assume the viability of MGTRMs would be about 14% based on flow cytometry results shown in Fig. 3a in the revised manuscript.

6) The presentation of data from 4T1 and E0071 should be consistent throughout the manuscript, not choose one or the other for supplemental.

Following the reviewer's comment, we have moved the E0771 data from the Supplementary Information to the main results.

7) The cytokine data in figure 2 is an afterthought; there is no follow-up to mechanisms. At least this should be explained in the discussion in the context of defined lineages of Balb/c and C57Bl6.

Thank you for the comment. We would like to remove the cytokine array results and the corresponding text from this manuscript. By further advancing the mechanism research, we would like to publish the results in a paper.

Reviewer #3 (Remarks to the Author):

This study aimed to investigate the role of resident macrophages in early TNBC development, recurrence, and metastases. The data are convincing; however, the paper still need to be revised.

Thank you for your valuable comments and the time you spent reviewing our manuscript. The revised major words and sentences are underlined in the text of the revised manuscript.

1) In Figure 2, I don't understand why the TNBC cell line was cultured separated from the macrophage since they are in close contacts in the tumor. This need to be clarified.

When elucidating cell-cell communication, investigating whether direct contact is required or mediated by humoral factors is the most common approach. In this study, we want to investigate cell-cell communication between TNBC cells and MGTRMs at the time of cancer development (or recurrence). At that time, the cells are not always in direct contact with each other, so according to the usual method, we first observed growth in culture conditions where cells do not come into direct contact with each other. In case of the experiment shown in Fig. 2a, we observed TNBC cell proliferation-promoting activity without cell-to-cell contact, concluding that cell-to-cell contact is not necessarily required to stimulate TNBC cell proliferation and that secreted components from mammary tissue (MGFP) are involved in the proliferation of TNBC cells.

In addition, by not adding mammary gland tissue fragments and PL/CL to the same well as TNBC cells, we were able to compare TNBC proliferation under the same culture conditions and obtain more reliable results.

We also confirmed that MGTRMs promoted TNBC cell growth even under cell-cell contact culture conditions in Fig. 2b.

2) Moreover, the authors used a TNBC cell line expressing the luciferase to evaluate its proliferation. To be able to use the luciferase activity as a readout for proliferation in vitro, the authors must show that the MGFP secretome, the CL and PL has no effect on the expression of the luciferase RNA in the TNBC cell line. A standard proliferation assay by counting the cell would be more appropriate.

As shown in Fig. 2a, it is clear that PL (PBS encapsulating liposome) treatment did not affect the luciferase activity of TNBC cells. In Supplementary Fig. 2c in the original manuscript (Fig. 2c in the revised manuscript), we compared luciferase activity in TNBC cells co-cultured with and without CL. There was no difference between them and thus no reason to suspect that CL (clodronate encapsulating liposome) could affect luciferase expression (transcription and

translation) and activity. To more directly present luciferase activity, the Y-axis designation of Fig. 2c in the revised manuscript was changed to Relative Light Units (RLU) instead of relative luciferase activity (the luciferase activity of TNBC cells normalized by the luciferase activity of corresponding TNBC cells without (-) CL.).

3) I feel that assessment of the cytokines secretion in this study is dispensable and does not bring any clues.

Thank you for the comment. We would like to remove the cytokine array results and the corresponding text from this manuscript, and advance the research to provide useful information in future publications.

Reviewers' comments:

Reviewer #1 (Remarks to the Author):

I am satisfied with the authors responses to most of my comments

One important last point should be addressed

Authors claim that all F4/80+ cells are macrophages but it is well established that F4/80 is also expressed by other phagocytes including monocytes and type 2 dendritic cells. We therefore do not know whether FOLR2-F480+ cells are macrophages or not. This point should be stated in their conclusion.

Reviewer #2 (Remarks to the Author):

The authors have addressed most of my concerns. Still, I think the technical and lineage-defining limitations of the study should be addressed clearly in the discussion.

Reviewer #3 (Remarks to the Author):

The authors adressed all of my concerns

Reviewer #1 (Remarks to the Author):

I am satisfied with the authors responses to most of my comments

One important last point should be addressed

Authors claim that all F4/80+ cells are macrophages but it is well established that F4/80 is also expressed by other phagocytes including monocytes and type 2 dendritic cells. We therefore do not know whether FOLR2-F480+ cells are macrophages or not. This point should be stated in their conclusion.

Thank you for your comment on our revised manuscript. I have added the following sentences to the discussion section (lines 225-231, underlined text) with 2 references (lines 581-585, underlined references):

Although many studies of MGTRMs including us use F4/80 as a macrophage marker, other cells such as monocytes and type 2 dendritic cells are known to express F4/80^{46, 47}. Intensive research on MGTRMs has just begun, and it is difficult to conclude from the current limited information that the F4/80⁺ population are entirely MGTRMs. Detailed studies on plasticity and heterogeneity of MGTRMs using multi-omics are expected to provide more relevant information of MGTRM subsets, leading to the promotion of research that leads to MGTRM-targeted therapy.

46. Crane, M. J. et al. The monocyte to macrophage transition in the murine sterile wound. *PLoS One*.

9, e86660 (2014).

47. Nguyen, M. T. A. et al. A subpopulation of macrophages infiltrates hypertrophic adipose tissue and is activated by free fatty acids via toll-like receptors 2 and 4 and JNK-dependent pathways. *J. Biol. Chem.* **282**, 35279–35292 (2007).

Reviewer #2 (Remarks to the Author):

The authors have addressed most of my concerns. Still, I think the technical and lineage-defining limitations of the study should be addressed clearly in the discussion.

Thank you for pointing out an important issue. I have added the following sentences to the discussion section (lines 212-215, underlined text):

However, lineage-definition remains difficult due to the lack of research on MGTRMs: Most of F4/80⁺ cells in MGFP are CD45-positive²⁵ and MGTRMs would have plasticity and heterogeneity that are hallmark features of macrophages as they can rapidly adjust their functional phenotype in response to their surrounding environment²².

Reviewer #3 (Remarks to the Author):

The authors addressed all of my concerns

Thank you for your effort and time to review our revised manuscript.